# Asymptotic Instance-Optimal Algorithms for Interactive Decision Making

**Kefan Dong**
Stanford University
kefandong@stanford.edu

**Tengyu Ma**
Stanford University
tengyuma@stanford.edu

## Abstract

Past research on interactive decision making problems (bandits, reinforcement learning, etc.) mostly focuses on the minimax regret that measures the algorithm's performance on the hardest instance. However, an ideal algorithm should adapt to the complexity of a particular problem instance and incur smaller regrets on easy instances than worst-case instances. In this paper, we design the first asymptotic instance-optimal algorithm for general interactive decision making problems with finite number of decisions under mild conditions. On *every* instance $f$, our algorithm outperforms *all* consistent algorithms (those achieving non-trivial regrets on all instances), and has asymptotic regret $\mathcal{C}(f) \ln n$, where $\mathcal{C}(f)$ is an exact characterization of the complexity of $f$. The key step of the algorithm involves hypothesis testing with active data collection. It computes the most economical decisions with which the algorithm collects observations to test whether an estimated instance is indeed correct; thus, the complexity $\mathcal{C}(f)$ is the minimum cost to test the instance $f$ against other instances. Our results, instantiated on concrete problems, recover the classical gap-dependent bounds for multi-armed bandits (Lai et al., 1985) and prior works on linear bandits (Lattimore & Szepesvari, 2017), and improve upon the previous best instance-dependent upper bound (Xu et al., 2021) for reinforcement learning.

## 1 Introduction

Bandit and reinforcement learning (RL) algorithms demonstrated a wide range of successful real-life applications (Silver et al., 2016; 2017; Mnih et al., 2013; Berner et al., 2019; Vinyals et al., 2019; Mnih et al., 2015; Degrave et al., 2022). Past works have theoretically studied the regret or sample complexity of various interactive decision making problems, such as contextual bandits, reinforcement learning (RL), partially observable Markov decision process (see Azar et al. (2017); Jin et al. (2018); Dong et al. (2021); Li et al. (2019); Agarwal et al. (2014); Foster & Rakhlin (2020); Jin et al. (2020), and references therein). Recently, Foster et al. (2021) present a unified algorithmic principle for achieving the minimax regret—the optimal regret for the worst-case problem instances.

However, minimax regret bounds do not necessarily always present a full picture of the statistical complexity of the problem. They characterize the complexity of the most difficult instances, but potentially many other instances are much easier. An ideal algorithm should adapt to the complexity of a particular instance and incur smaller regrets on easy instances than the worst-case instances. Thus, an ideal regret bound should be instance-dependent, that is, depending on some properties of each instance. Prior works designed algorithms with instance-dependent regret bounds that are stronger than minimax regret bounds, but they are often not directly comparable because they depend on different properties of the instances, such as the gap conditions and the variance of the value function (Zanette & Brunskill, 2019; Xu et al., 2021; Foster et al., 2020; Tirinzoni et al., 2021).

A more ambitious and challenging goal is to design *instance-optimal* algorithms that outperform, on *every* instance, *all* consistent algorithms (those achieving non-trivial regrets on all instances). Past works designed instance-optimal algorithms for multi-armed bandit (Lai et al., 1985), linear bandits (Kirschner et al., 2021; Hao et al., 2020), Lipschitz bandits (Magureanu et al., 2014), and ergodic MDPs (Ok et al., 2018). However, instance-optimal regret bounds for tabular reinforcement learning remain an open question, despite recent progress (Tirinzoni et al., 2021; 2022). The challenge partly

stems from the fact that the *existence* of such an instance-optimal algorithm is even a priori unclear for general interactive decision making problems. Conceivably, each algorithm can have its own specialty on a subset of instances, and no algorithm can dominate all others on all instances.

Somewhat surprisingly, we prove that there exists a simple algorithm (T2C, stated in Alg. 1) that is asymptotic instance-optimal for general interactive decision making problems with finite number of decisions.We determine the *exact* leading term of the optimal asymptotic regret for instance $f$ to be $\mathcal{C}(f)\ln n$. Here, $n$ is the number of interactions and $\mathcal{C}(f)$ is a complexity measure for the instance $f$ that intuitively captures the difficulty of distinguishing $f$ from other instances (that have different optimal decisions) using observations. Concretely, under mild conditions on the interactive decision problem, our algorithm achieves an asymptotic regret $\mathcal{C}(f)\ln n$ (Theorem 5.2) for every instance $f$, while every consistent algorithm must have an asymptotic regret at least $\mathcal{C}(f)\ln n$ (Theorem 3.2).

Our algorithm consists of three simple steps. First, it explores uniformly for $o(1)$-fraction of the steps and computes the MLE estimate of the instance with relatively low confidence. Then, it tests whether the estimate instance (or, precisely, its associated optimal decision) is indeed correct using the most economical set of queries/decisions. Concretely, it computes a set of decisions with minimal regret, such that, using a log-likelihood ratio test, it can either distinguish the estimated instance from all other instances (with different optimal decisions) with high confidence, or determine that our estimation was incorrect. Finally, with the high-confidence estimate, it commits to the optimal decision of the estimated instance in the rest of the steps. The algorithmic framework essentially reduces the problem to the key second step — optimal hypothesis testing with active data collection.

Our results recover the classical gap-dependent regret bounds for multi-armed bandits (Lai et al., 1985) and prior works on linear bandits (Lattimore & Szepesvari, 2017; Hao et al., 2020). As an instantiation of the general algorithm, we present the first asymptotic instance-optimal algorithm for tabular RL, improving upon prior instance-dependent algorithms (Xu et al., 2021; Simchowitz & Jamieson, 2019; Tirinzoni et al., 2021; 2022).

## 1.1 ADDITIONAL RELATED WORKS

Some algorithms are proved instance-optimal for specific interactive decision making problems. Variants of UCB algorithms are instance-optimal for bandits with various assumptions (Lattimore & Szepesvári, 2020; Gupta et al., 2021; Tirinzoni et al., 2020; Degenne et al., 2020; Magureanu et al., 2014), but are suboptimal for linear bandits (Lattimore & Szepesvari, 2017). These algorithms rely on the optimism-in-face-of-uncertainty principle to deal with exploration-exploitation tradeoff, whereas our algorithm explicitly computes the best tradeoff. Kirschner et al. (2021); Lattimore & Szepesvari (2017); Hao et al. (2020) design non-optimistic instance-optimal algorithms for linear bandits. There are also instance-optimal algorithms for ergodic MDPs where the regret is less sensitive to the exploration policy (Ok et al., 2018; Burnetas & Katehakis, 1997; Graves & Lai, 1997), interactive decision making with finite hypothesis class, finite state-action space, and known rewards (Rajeev et al., 1989), and interactive decision making with finite observations (Komiyama et al., 2015).

The most related problem setup is structured bandits (Combes et al., 2017; Van Parys & Golrezaei, 2020; Jun & Zhang, 2020), where the instances also belong to an abstract and arbitrary family $\mathcal{F}$. The structured bandits problem is a very special case of general decision making problems and does not contain RL because the observation is a scalar. In contrast, the observation in general decision making problems could be high-dimensional (e.g., a trajectory with multiple states, actions, and rewards for episodic RL), which makes our results technically challenging.

Many algorithms' regret bounds depend on some properties of instances such as the gap condition. Foster et al. (2020) prove a gap-dependent regret bound for contextual bandits. For reinforcement learning, the regret bound may depend on the variance of the optimal value function (Zanette & Brunskill, 2019) or the gap of the $Q$-function (Xu et al., 2021; Simchowitz & Jamieson, 2019; Yang et al., 2021). Xu et al. (2021); Foster et al. (2020) also prove that the gap-dependent bounds cannot be improve on some instances. To some extent, these instance-dependent lower bounds can be viewed as minimax bounds for a fine-grained instance family (e.g., all instances with the same $Q$-function gap), and therefore are different from ours.

## 2 SETUP AND NOTATIONS

**Interactive decision making problems.** We focus on interactive decision making problem with structured observations (Foster et al., 2021), which includes bandits and reinforcement learning as special cases. An interactive decision making problem is defined by a space of decisions $\Pi$, observations $O$, a reward function $R : O \to \mathbb{R}$, and a function $f$ (also called an instance) that maps a decision $\pi \in \Pi$ to a distribution over observations $f[\pi]$. We use $f[\pi](\cdot) : O \to \mathbb{R}_+$ to denote the density function of the distribution $f[\pi]$. We assume that the reward $R$ is a deterministic and known.

An environment picks an ground-truth instance $f^\star$ from a instance family $\mathcal{F}$, and then an agent (which knows the instance family $\mathcal{F}$ but not the ground-truth $f^\star$) interacts with the environment for $n$ total rounds. In round $t \le n$,

  (a) the learner selects a decision $\pi_t$ from the decision class $\Pi$, and
  (b) the environment generates an observation $o_t$ following the ground-truth distribution $f^\star[\pi_t]$ and reveals the observation. Then the agent receives a reward $R(o_t)$.

For example, multi-armed bandits, linear bandits, and *episodic* reinforcement learning all belong to interactive decision making problems. For bandits, a decision $\pi$ corresponds to an action and an observation $o$ corresponds to a reward. For reinforcement learning, a decision $\pi$ is a deterministic policy that maps from states to actions, and an observation $o$ is a trajectory (including the reward at each step). In other words, one round of interactions in the interactive decision making problem corresponds to one episode of the RL problem. We will formally discuss the RL setup in Section 5.

Let $R_f(\pi) = \mathbb{E}_{o \sim f[\pi]}[R(o)]$ be the expected reward for decision $\pi$ under instance $f$, and $\pi^\star(f) \triangleq \arg\max_\pi R_f(\pi)$ the optimal decision of instance $f$. The expected regret measures how much worse the agent's decision is than the optimal decision:

$$\mathrm{Reg}_{f,n} = \mathbb{E}_f \left[ \sum_{t=1}^n \left( \max_{\pi \in \Pi} R_f(\pi) - R_f(\pi_t) \right) \right]. \tag{1}$$

We consider the case where the decision family $\Pi$ is *finite*, and every instance $f \in \mathcal{F}$ has a unique optimal decision, denoted by $\pi^\star(f)$. We also assume that for every $f \in \mathcal{F}$ and $\pi \in \Pi$, $0 \le R(o) \le R_{\max}$ almost surely when $o$ is drawn from the distribution $f[\pi]$, and $\sup_o f[\pi](o) < \infty$.

**Consistent algorithms and asymptotic instance-optimality.** We first note that it's unreasonable to ask for an algorithm that can outperform or match any *arbitrary* algorithm on *every* instance. This is because, for any instance $f \in \mathcal{F}$, a bespoke algorithm that always outputs $\pi^\star(f)$ can achieve zero regret on instance $f$ (though it has terrible regrets for other instances). Therefore, if an algorithm can outperform or match any algorithm on any instance, it must have zero regret on every instance, which is generally impossible. Instead, we are interested in finding an algorithm that is as good as any other reasonable algorithms that are not completely customized to a single instance.

We say an algorithm is consistent if its expected regret satisfies $\mathrm{Reg}_{f,n} = o(n^p)$ for *every* $p > 0$ and $f \in \mathcal{F}$ (Lai et al., 1985). Most of the reasonable algorithms are consistent, such as the UCB algorithm for multi-armed bandits (Lattimore & Szepesvári, 2020), UCBVI, and UCB-Q algorithm for tabular reinforcement learning (Simchowitz & Jamieson, 2019; Yang et al., 2021), because all of them achieve asymptotically $\mathcal{O}(\ln n)$ regrets on any instance, where $\mathcal{O}$ hides constants that depend on the property of the particular instance.[1] However, consistent algorithms exclude the algorithm mentioned in the previous paragraph which is purely customized to a particular instance.

We say an algorithm is asymptotically *instance-optimal* if on *every* instance $f \in \mathcal{F}$, $\limsup_{n \to \infty} \mathrm{Reg}_{f,n} / \ln n$ is the smallest among all *consistent* algorithms (Lai et al., 1985). We note that even though an instance-optimal algorithm only needs to perform as well as every consistent algorithm, a priori, it's still unclear if such an algorithm exists.

Some prior works have also used a slightly weaker definition in place of consistent algorithms, e.g., $\alpha$-uniformly good algorithms in Tirinzoni et al. (2021), which allows a sublinear regret bound $\mathcal{O}(n^\alpha)$ for *some* constant $\alpha < 1$. The alternative definition, though apparently includes more algorithms, does not change the essence. Our algorithm is still instance-optimal *up to a constant factor*— simple modification of the lower bound part of the proof shows that its asymptotic regret is at most

---

[1]Here the regret scales in $\log n$ because we are in the asymptotic setting where the instance is fixed and $n$ tends to infinity. If the instance can depend on $n$ (which is not the setting we are interested in), then the minimax regret typically scales in $O(\sqrt{n})$.

$(1 - \alpha)^{-1}$ factor bigger than any $\alpha$-uniformly good algorithms on any instance. This paper thus primarily compares with consistent algorithms for simplicity.

## 3    MAIN RESULTS

In this section, we present an intrinsic complexity measure $\mathcal{C}(f)$ of an instance $f$ (Section 3.1), and an instance-optimal algorithm that achieves an asymptotic regret $\mathcal{C}(f) \ln n$ (Section 3.2).

### 3.1    COMPLEXITY MEASURE AND REGRET LOWER BOUNDS

In this section, our goal is to understand the minimum regret of consistent algorithms, which is an intrinsic complexity measure of an instance. We define an instance-dependent complexity measure $\mathcal{C}(f)$ and prove that any consistent algorithm must have asymptotic regret at least $\mathcal{C}(f) \ln n$.

The key observation is that any consistent algorithm has to collect enough information from the observations to tell apart the ground-truth instance from other instances with different optimal decisions. Consider the situation when a sequence of $n$ decisions is insufficient to distinguish two instances, denoted by $f$ and $g$, with different optimal decisions. If an algorithm achieves a sublinear regret on $f$, the sequence must contain $\pi^\star(f)$ (the optimal decision for $f$) at least $n - o(n)$ times. As a result, if the true instance were $g$, the algorithm suffers a linear regret (due to the linear number of $\pi^\star(f)$), and therefore cannot be consistent.

An ideal algorithm should find out decisions that can *most efficiently* identify the ground-truth instance (or precisely the family of instances with the same optimal decision as the ground-truth). However, decisions collect information but also incur regrets. So the algorithm should pick a list of decisions with the best tradeoff between these two effects—minimizing the decisions' regret and collecting sufficient information to identify the true instance. Concretely, suppose a sequence of decisions includes $w_\pi$ occurrences of decision $\pi$. The regret of these decisions is $\sum_{\pi \in \Pi} w_\pi \Delta(f, \pi)$, where $\Delta(f, \pi) \triangleq R_f(\pi^\star(f)) - R_f(\pi)$ is the sub-optimality gap of decision $\pi$. We use KL divergence between the observations of $\pi$ under two instances $f$ and $g$ (denoted by $D_{\mathrm{KL}}(f[\pi] \| g[\pi])$) to measure $\pi$'s power to distinguish them. The following optimization problem defines the optimal mixture of the decisions (in terms of the regret) that has sufficient power to distinguish $f$ from all other instances with different optimal decision.

$$\mathcal{C}(f, n) \triangleq \min_{w \in \mathbb{R}_+^{|\Pi|}} \sum_{\pi \in \Pi} w_\pi \Delta(f, \pi) \tag{2}$$

$$\text{s.t.} \quad \sum_{\pi \in \Pi} w_\pi D_{\mathrm{KL}}(f[\pi] \| g[\pi]) \geq 1, \ \forall g \in \mathcal{F}, \pi^\star(g) \neq \pi^\star(f), \tag{3}$$

$$\|w\|_\infty \leq n. \tag{4}$$

The last constraint makes sure that $w_\pi < \infty$ even if the decision has no regret (i.e., $\exists \pi, \Delta(f, \pi) = 0$). We only care about the case when $n$ approaches infinity. The asymptotic complexity of $f$ is

$$\mathcal{C}(f) \triangleq \lim_{n \to \infty} \mathcal{C}(f, n). \tag{5}$$

In Eq. (3), we only require separation between instances $f, g$ when they have different optimal decisions. As there are only finite decisions, there are finite equivalent classes, so $f$ and $g$ are in principle separable with sufficient number of observations. Thus, the complexity measure $\mathcal{C}(f, n)$ is well defined as stated in the following lemma.

**Lemma 3.1.** *For any $f \in \mathcal{F}$, $\mathcal{C}(f, n)$ is non-increasing in $n$, and there exists $n_0 > 0$ such that for all $n > n_0$, $\mathcal{C}(f, n) < \infty$. As a corollary, $\mathcal{C}(f) < \infty$ and is well defined.*

Proof of Lemma 3.1 is deferred to Appendix B.1. In Section 3.1, we formalize the intuition above and prove that $\mathcal{C}(f)$ is a lower bound of the asymptotic regret, as stated in the following theorem.

**Theorem 3.2.** *For every instance $f \in \mathcal{F}$, the expected regret of any consistent algorithm satisfies* $\limsup_{n \to \infty} \frac{\mathrm{Reg}_{f,n}}{\ln n} \geq \mathcal{C}(f)$.

We prove Theorem 3.2 in Appendix B.2. Theorem 3.2 is inspired by previous works (Lattimore & Szepesvari, 2017; Tirinzoni et al., 2021). When applied to tabular RL problems, our lower bound is very similar to that in Tirinzoni et al. (2021) with the only difference that their optimization problems omit the second constraint (Eq. (4)), which can make it slightly less tight (see Section B.5).

The complexity measure $\mathcal{C}(f)$ reduces to the well-known inverse action gap bound $\mathcal{O}(\sum_{a \in \mathcal{A}, \Delta(a)>0} 1/\Delta(a))$ for multi-armed bandits (Proposition B.2), and recovers the result of Lattimore & Szepesvari (2017) for linear bandits (Proposition B.3). For reinforcement learning, Tirinzoni et al. (2021) prove that the instance-optimal bound can be smaller than the gap-dependent bound $\mathcal{O}(\sum_{s,a:\Delta(s,a)>0} 1/\Delta(s,a))$ in Xu et al. (2021) and Simchowitz & Jamieson (2019).

## 3.2 Instance-Optimal Algorithms

---

**Algorithm 1** Test-to-Commit (T2C)

---

1: Parameters: the number of rounds of interactions $n$.

**Step 1: Initialization.**

2: Play each decision $\lceil \frac{\ln n}{\ln \ln n} \rceil$ times. We denote these decisions by $\{\hat{\pi}_i\}_{i=1}^{m_{\text{init}}}$, where $m_{\text{init}} = |\Pi| \lceil \frac{\ln n}{\ln \ln n} \rceil$, and the corresponding observations by $\{\hat{o}_i\}_{i=1}^{m_{\text{init}}}$.

3: Compute the max likelihood estimation (MLE) with arbitrary tie-breaking

$$\hat{f} = \text{argmax}_{f \in \mathcal{F}} \ \sum_{i=1}^{m_{\text{init}}} \ln f[\hat{\pi}_i](\hat{o}_i). \tag{6}$$

**Step 2: Identification.**

4: Let $\eta = (\ln \ln n)^{1/4}$ for shorthand, and $\hat{w}$ be the solution of the program defining $\mathcal{C}(\hat{f}, \eta)$. Compute $\bar{w}_\pi = (1 + 1/\eta)\hat{w}_\pi + 1/\eta$ for all $\pi \in \Pi$.

5: Play each decision $\pi$ for $\lceil \bar{w}_\pi \ln n \rceil$ times. Denote these decisions by $w = \{\pi_i\}_{i=1}^{m}$ where $m = \sum_\pi \lceil \bar{w}_\pi \ln(n) \rceil$, and the corresponding observations by $\{o_i\}_{i=1}^{m}$.

6: Run the log-likelihood ratio test on instance $\hat{f}$, the sequence of decision $\{\pi_i\}_{i=1}^{m}$ and its corresponding observations $\{o_i\}_{i=1}^{m}$ (that is, compute the event $\mathcal{E}_{\text{acc}}^{\hat{f}}$ defined in Eq. (8)).

**Step 3: Exploitation.**

7: **if** $\mathcal{E}_{\text{acc}}^{\hat{f}} = \text{true}$ **then**

8:     Commit to $\pi^\star(\hat{f})$ (i.e., run $\pi^\star(\hat{f})$ for the remaining steps).

9: **else**

10:     Run UCB for the remaining steps.

---

We first present the regret bound for our algorithm of the T2C algorithm. For simplicity, we consider a finite hypothesis (i.e., $|\mathcal{F}| < \infty$) here, and extend to infinite hypothesis case in Section 5.

We start by stating a condition that excludes abnormal observation distributions $f[\pi]$. Recall that for any $\zeta \in (0, 1)$, the Rényi divergence of two distributions $p, q$ is

$$D_\zeta(p\|q) = \tfrac{1}{\zeta-1} \ln \int_x p(x)^\zeta q(x)^{1-\zeta} \mathrm{d}x. \tag{7}$$

The Rényi divergence $D_\zeta(p\|q)$ is non-decreasing in $\zeta$, and $\lim_{\zeta \uparrow 1} D_\zeta(p\|q) = D_{\text{KL}}(p\|q)$ (Van Erven & Harremos, 2014). We require the limits converge uniformly for all instances $g \in \mathcal{F}$ and decisions $\pi \in \Pi$ with a $\zeta$ bounded away from 1 (the choice of the constants in Condition 1 is not critical to our result), as stated below.

**Condition 1.** *For any fixed $\alpha > 0, \epsilon > 0$, instance $f \in \mathcal{F}$, there exists $\lambda_0(\alpha, \epsilon, f) > 0$ such that for all $\lambda \leq \lambda_0(\alpha, \epsilon, f)$, $g \in \mathcal{F}$ and $\pi \in \Pi$, $D_{1-\lambda}(f[\pi]\|g[\pi]) \geq \min\{D_{\text{KL}}(f[\pi]\|g[\pi]) - \epsilon, \alpha\}$. Moreover, we require $\lambda_0(\alpha, \epsilon, f) \geq \epsilon^{c_1} \min\{1/\alpha, c_2\}^{c_3} \iota(f)$ for some universal constants $c_1, c_2, c_3 > 0$, where $\iota(f) > 0$ is a function that only depends on $f$.*

Condition 1 holds for a wide range of distributions, such as Gaussian, Bernoulli, multinomial, Laplace with bounded mean, Log-normal (Gil et al., 2013), and tabular RL where $f[\pi]$ is a distribution over a trajectory consists of state, action and reward tuples (see Theorem 5.1 for the proof of tabular RL). A stronger but more interpretable variant of Condition 1 is that the log density ratio of $f[\pi]$ and $g[\pi]$ has finite fourth moments (see Proposition B.4), therefore Condition 1 can also be potentially verified for other distributions.

The main theorem analyzing Alg. 1 is shown below. The asymptotic regret of Alg. 1 matches the constants in the lower bound (Theorem 3.2), indicating its asymptotic instance-optimality.

**Theorem 3.3.** *Suppose $\mathcal{F}$ is a finite hypothesis class and satisfies Condition 1 The regret of Alg. 1 satisfies $\limsup_{n \to \infty} \frac{\text{Reg}_{f^\star, n}}{\ln n} \leq \mathcal{C}(f^\star)$.*

Our algorithm is stated in Alg. 1 and consists of three steps: initialization, identification, and exploitation. In the initialization step, we explore uniformly for a short period and compute the MLE estimate $\hat{f}$ of the true instance (Line 3 of Alg. 1), where we only requires $\hat{f}$ to be accurate with moderate probability (i.e., $1 - 1/\ln n$). The estimation is used to compute the lowest-regret list of decisions that can distinguish the optimal decision of $\hat{f}$. Since we only collect $o(\ln n)$ samples, the regret of this step is negligible asymptotically.

In the identification step, we hold the belief that $\hat{f}$ is the true instance and solve $\mathcal{C}(\hat{f}, (\ln \ln n)^{1/4})$ to get the best list of decisions to fortify this belief (see Line 4 of Alg. 1). Then we collect more samples using this list (with minor decorations) to boost the confidence of our initial estimation to $1 - 1/n$ (or reject it when the estimation is not accurate) by the following log-likelihood ratio test:

$$\mathcal{E}_{\text{acc}}^{\hat{f}} = \mathbb{I}\big[\forall g \in \mathcal{F} \text{ and } \pi^\star(g) \neq \pi^\star(\hat{f}), \sum_{i=1}^{m} \ln \frac{\hat{f}[\pi_i](o_i)}{g[\pi_i](o_i)} \geq \ln n\big]. \tag{8}$$

Intuitively, if $\hat{f}$ is not the ground-truth $f^\star$, $\mathcal{E}_{\text{acc}}^{\hat{f}}$ is unlikely to hold because the expected log-likelihood ratio is non-positive: $\mathbb{E}_{o \sim f^\star[\pi]}[\ln(\hat{f}[\pi](o)/f^\star[\pi](o))] = -D_{\text{KL}}(f^\star[\pi] \| \hat{f}[\pi]) \leq 0$. Hence, with high probability a wrong guess cannot be accepted. On the other hand, the first constraint (Eq. (3)) in the definition of $\mathcal{C}(\hat{f})$ guarantees that when $\hat{f} = f^\star$, the expected log-likelihood ratio is large for all $g \in \mathcal{F}$ and $\pi^\star(g) \neq \pi^\star(\hat{f})$, so an accurate guess will be accepted. In this step $m = \tilde{\Theta}(\ln n)$, so Step 2 dominates the regret of Alg. 1. In other words, the efficiency of the log-likelihood test is critical to our analysis.

Finally in the exploitation step, the algorithm commits to the optimal decision of $\hat{f}$ if it believes this estimation with confidence $1 - 1/n$, or run a standard UCB algorithm as if on a multi-armed bandits problem (with a well-known $\mathcal{O}(\ln n)$ regret) when the initial estimation is not accurate (happens with probability at most $1/\ln n$). Hence, the expected regret here is $\mathcal{O}(\ln n)/\ln n + \mathcal{O}(n)/n = \mathcal{O}(1)$.

Our three-stage algorithm is inspired by Lattimore & Szepesvari (2017), where their test in Step 2 only uses the reward information. However, the observation for general interactive decision making problem contains much more information such as the transition in tabular RL problems. In contrast, the T2C algorithm relies on a general hypothesis testing method (i.e., log-likelihood ratio test) and achieves optimal regret for general interactive decision making problems.

## 4    PROOF SKETCHES OF THE MAIN RESULTS

In this section, we discuss the proof sketch of our main results. Section 4.1 discusses the proof sketch of the lower bound, and Section 4.2 shows the main lemmas for each step in Alg. 1. In Section 4.3, we discuss the log-likelihood ratio test in detail.

### 4.1    PROOF SKETCH OF THEOREM 3.2

Recall that $\mathcal{C}(f)$ is the minimum regret of a list of decisions that can distinguish the instance $f$ with all other instances $g$ (with a different optimal decision). Hence, to prove the lower bound, we show that the sequence of decisions played by any consistent algorithm must also distinguish $f$, and thus, $\mathcal{C}(f)$ lower bounds the regret of any consistent algorithm.

For any consistent algorithm, number of interactions $n > 0$, and two instances $f, g \in \mathcal{F}$ with different optimal decisions, let $P_{f,n}$ and $P_{g,n}$ denote the probability space generated by running the algorithm on instances $f$ and $g$ respectively for $n$ rounds. Since $f, g$ have different optimal decisions, $P_{f,n}$ and $P_{g,n}$ should be very different — following the same proof strategy in (Lattimore & Szepesvari, 2017), we can show that

$$D_{\text{KL}}(P_{f,n} \| P_{g,n}) \geq (1 + o(1)) \ln n.$$

Let the random variable $\pi_i$ be the decision of the algorithm in round $i$. The chain rule of KL divergence shows

$$D_{\text{KL}}(P_{f,n} \| P_{g,n}) = \mathbb{E}_{f,n}\big[\sum_{i=1}^{n} D_{\text{KL}}(f[\pi_i] \| g[\pi_i])\big] = \sum_{\pi} \mathbb{E}_{f,n}[N_\pi] D_{\text{KL}}(f[\pi] \| g[\pi]).$$

Now consider the vector $w \in \mathbb{R}_+^{|\Pi|}$ where $w_\pi = \mathbb{E}_f[N_\pi]/((1 + o(1)) \ln n)$. Based on the derivation above, we can verify that $w$ is a valid solution to $\mathcal{C}(f, n)$, and therefore

$$\text{Reg}_{f,n} = \sum_{\pi} \mathbb{E}_f[N_\pi] \Delta(f, \pi) = \sum_{\pi} w_\pi \Delta(f, \pi)(1 + o(1)) \ln n \geq \mathcal{C}(f, n)(1 + o(1)) \ln n.$$

Then the final result is proved by the fact that $\mathcal{C}(f) = \lim_{n \to \infty} \mathcal{C}(f, n)$.

## 4.2 Proof Sketch of Theorem 3.3

In the following, we discuss main lemmas and their proof sketches for the three steps of Alg. 1.

**Step 1: Initialization.** In this step we show that the max likelihood estimation can find the exact instance (i.e., $\hat{f} = f^\star$) with probability at least $1 - 1/\ln n$ and negligible regret. Note that the failure probability is not small enough to directly commit to the optimal decision of $\hat{f}$ (i.e., play $\pi^\star(\hat{f})$ forever), which would incur linear regret when $\hat{f} \neq f^\star$. Formally speaking, the main lemma for Step 1 is stated as follows, whose proof is deterred to Appendix A.1.

**Lemma 4.1.** *Under Condition 1, with probability at least $1 - 1/\ln n$ we get $\hat{f} = f^\star$. In addition, the regret of Step 1 is upper bounded by $\mathcal{O}(\frac{\ln n}{\ln \ln n})$.*

Lemma 4.1 is not surprising since the MLE only requires $\mathcal{O}(\log(1/\delta))$ samples to reach a failure probability $\delta$ in general (Van de Geer, 2000, Section 7). Here we require $1/\ln n$ failure probability but allow $\ln n/(\ln \ln n) = \Omega(\ln \ln n)$ samples, hence the result is expected for large enough $n$.

**Step 2: Identification.** In the identification step, we boost the failure probability to $1/n$ using the log-likelihood test. To this end, we compute the optimal list of decisions $w = \{\pi_1, \cdots, \pi_m\}$ that distinguishes $\hat{f}$ by solving $\mathcal{C}(\hat{f}, (\ln \ln n)^{1/4})$ (Line 4 of Alg. 1). The choice of $(\ln \ln n)^{1/4}$ is not critical, and could be replaced by any smaller quantity that approaches infinity as $n \to \infty$. Then we run the log-likelihood ratio test using the observations collected by executing the list of decision $w$, and achieve the following:

(a) when the true instance $f^\star$ and the estimation $\hat{f}$ have different optimal decisions, accept $\hat{f}$ with probability at most $1/n$;

(b) when $\hat{f} = f^\star$, accept $\hat{f}$ with probability at least $1 - 1/\ln n$.

In Section 4.3, we will discuss the log-likelihood ratio test in detail. Note that the regret after we falsely reject the true instance is $\mathcal{O}(\ln n)$ (by the regret bound of UCB algorithm), so we only require a $1/\ln n$ failure probability for (b) because then it leads to a $\mathcal{O}(\ln n)/\ln n = \mathcal{O}(1)$ expected regret. The main lemma for Step 2 is stated as follows, and its proof is deferred to Appendix A.2

**Lemma 4.2.** *Under Condition 1, for any finite hypothesis $\mathcal{F}$, for large enough $n$ the following holds:*

(a) *conditioned on the event $\pi^\star(\hat{f}) \neq \pi^\star(f^\star)$, $\mathcal{E}_{\mathrm{acc}}^{\hat{f}}$ is true with probability at most $1/n$;*

(b) *conditioned on the event $\hat{f} = f^\star$, $\mathcal{E}_{\mathrm{acc}}^{\hat{f}}$ is true with probability at least $1 - 1/\ln n$;*

(c) *conditioned on the event $\hat{f} = f^\star$, the expected regret of Step 2 is upper bounded by $\left(\mathcal{C}(f^\star, (\ln \ln n)^{1/4}) + o(1)\right) \ln n$. Otherwise, the expected regret of Step 2 is upper bounded by $\mathcal{O}(\ln n \ln \ln n)$.*

**Step 3: Exploitation.** Finally, in Step 3 we either commits to the optimal decision $\pi^\star(\hat{f})$ when the estimation $\hat{f}$ is accepted, or run a standard UCB algorithm with $\mathcal{O}(\ln n)$ regret (Lattimore & Szepesvári, 2020). Combining Lemma 4.1 and Lemma 4.2 we can prove that

(a) the probability of Step 3 incurring a $\mathcal{O}(n)$ regret is at most $1/n$, and

(b) the probability of Step 3 incurring a $\mathcal{O}(\ln n)$ regret is at most $1/\ln n$.

As a result, the expected regret in Step 3 is $\mathcal{O}(1)$ and negligible . Finally Theorem 3.3 is proved by stitching the three steps together, and deferred to Appendix A.4.

## 4.3 The Log-Likelihood Ratio Test

The goal of the log-likelihood ratio test is to boost the confidence of our initial estimation $\hat{f}$ to $1 - 1/n$, so that the algorithm can safely commit to its optimal decision in Step 3. In other words, the test should reject a wrong estimation but also accept a correct one.

Formally speaking, in the test we observe a list of observations $\{o_1, \cdots, o_m\}$ collected by the list of decision $\{\pi_1, \cdots, \pi_m\}$ on the *true instance* $f^\star$, and achieve the following two goals simultaneously,

(a) when $\hat{f}$ and $f^\star$ are sufficiently different (i.e., their KL divergence is large in the sense that $\sum_{i=1}^m D_{\mathrm{KL}}(\hat{f}[\pi_i] \| f^\star[\pi_i]) \geq (1 + o(1)) \ln n$), accept $\hat{f}$ with probability at most $1/n$, and

(b) when $\hat{f} = f^\star$, accept $\hat{f}$ with probability at least $1 - 1/\ln n$.

We prove that the log-likelihood ratio test with proper parameters achieves (a) and (b) under Condition 1 in the next lemma, whose proof is deferred to Appendix A.3.

**Lemma 4.3.** *Given two sequences of distributions $P = \{P_i\}_{i=1}^m$ and $Q = \{Q_i\}_{i=1}^m$, and a sequence of independent random variables $o = \{o_i\}_{i=1}^m$. For any fixed $\lambda > 0, c > 0$, and $\beta = \frac{1}{m} \sum_{i=1}^m D_{1-\lambda}(P_i \| Q_i)$, the test event $\mathcal{E}_{\mathrm{acc}} = \mathbb{I}\big[\sum_{i=1}^m \ln \frac{P_i(o_i)}{Q_i(o_i)} \geq c\big]$ satisfies*

$$\Pr_{o \sim Q}(\mathcal{E}_{\mathrm{acc}}) \leq \exp(-c), \tag{9}$$
$$\Pr_{o \sim P}(\mathcal{E}_{\mathrm{acc}}) \geq 1 - \exp(-\lambda(m\beta - c)). \tag{10}$$

To use Lemma 4.3 in the proof of Lemma C.2, we set $c = \ln n$ and $\lambda = \mathrm{poly}(\ln \ln n)$. Note that the choice of $w$ in Line 4 of Alg. 1 and Condition 1 implies $m\beta = \sum_{i=1}^m D_{1-\lambda}(P_i \| Q_i) \gtrapprox \sum_{i=1}^m D_{\mathrm{KL}}(\hat{f}[\pi_i] \| f^\star[\pi_i]) = (1 + o(1)) \ln n$ when $\hat{f}, f^\star$ have different optimal policies. So the conclusion of this lemma matches the first two items in Lemma 4.2.

Lemma 4.3 is closely related to the Chernoff-Stein lemma (see Chernoff (1959) and Mao (2021, Theorem 4.11)), with the difference that the failure probability of Eq. (10) in classical Chernoff-Stein lemma is a constant, while here it decreases with $m$. The proof of Eq. (9) is the same as in Chernoff-Stein lemma, and proof of Eq. (10) only requires an one-sided concentration of the empirical log-likelihood ratio. Indeed, the expectation of empirical log-likelihood ratio can be lower bounded by $\mathbb{E}_P\big[\sum_{i=1}^m \ln \frac{P_i(o_i)}{Q_i(o_i)}\big] = \sum_{i=1}^m D_{\mathrm{KL}}(P_i \| Q_i) \geq \sum_{i=1}^m D_{1-\lambda}(P_i \| Q_i) = m\beta$. Hence, $\Pr_P(\mathcal{E}_{\mathrm{acc}})$ is the probability that the empirical log-likelihood ratio is (approximately) larger than its expectation. We also note that the concentration is non-trivial because we do not make assumptions on the boundedness on the tail of $Q_i$.

## 5 Extensions to Infinite Hypothesis Class

Now we extend our analysis to infinite hypothesis settings, and instantiate our results on tabular RL. Our results in the infinite hypothesis case need two additional conditions. The first condition requires an upper bound on covering number of the hypothesis, and is used to prove a infinite-hypothesis version of Lemma 4.2. Formally speaking, for any $f, g \in \mathcal{F}$, define their distance as

$$d(f, g) = \sup_{\pi \in \Pi, o} |f[\pi](o) - g[\pi](o)|. \tag{11}$$

An $\epsilon$ (proper) covering of $\mathcal{F}$ is a subset $\mathcal{C} \subseteq \mathcal{F}$, such that for any $g \in \mathcal{F}$, there exists $g' \in \mathcal{C}$ with $d(g, g') \leq \epsilon$. The covering number $\mathcal{N}(\mathcal{F}, \epsilon)$ is the size of the minimum $\epsilon$ covering of $\mathcal{F}$.

**Condition 2.** *There exists constant $c$ that depends on $\mathcal{F}$ such that $\ln \mathcal{N}(\mathcal{F}, \epsilon) \leq \mathcal{O}(c \ln(1/\epsilon))$ for every $\epsilon > 0$. In addition, the base measure of the probability space has a finite volume $\mathrm{Vol} < \infty$.*

The second condition upper bounds the distance of two instances by a polynomial of their KL divergence, and is used to prove a stronger version of Lemma 4.1.

**Condition 3.** *There exists a constant $c_{\min} > 0$ (which may depend on $f^\star$) such that for all $\pi \in \Pi$ and $o \in \mathrm{supp}(f^\star[\pi])$ $f^\star[\pi](o) > c_{\min}$. In addition, there exists a constant $\iota(f^\star) > 0$ that only depends on $f^\star$ and $c_5 > 0$, such that for all $f \in \mathcal{F}, \pi \in \Pi$, when $D_{\mathrm{KL}}(f^\star[\pi] \| f[\pi]) \leq 1$*

$$\|f^\star[\pi] - f[\pi]\|_\infty \leq \iota(f^\star) D_{\mathrm{KL}}(f^\star[\pi] \| f[\pi])^{c_5}.$$

A lot of interactive decision making problems satisfy Conditions 2, 3, such as multi-armed bandits and linear bandits with Bernoulli reward or truncated Gaussian reward. For bandit problems, Conditions 2 and 3 only depend on the noise of the reward function and can be verified easily.

For tabular RL, an observation $o$ denotes a trajectory $(s_1, a_1, r_1, \cdots, s_H, a_H, r_H)$ where the state-action pairs $(s_h, a_h)$ are discrete random variables but the rewards $r_h$ are continuous. A decision

$\pi : \mathcal{S} \to \mathcal{A}$ is a mapping from the state space to action space, so $|\Pi| = |\mathcal{A}|^{|\mathcal{S}|} < \infty$. In Appendix D, we formally define the tabular RL problems where the reward $r_h$ given $(s_h, a_h)$ follows a truncated Gaussian distribution, and prove that Conditions 1-3 are satisfied as stated in the following theorem.

**Theorem 5.1.** *Let $\mathcal{F}$ be the family of tabular RL with truncated Gaussian reward and unique optimal policies. Then Conditions 1, 2, 3 holds simultaneously.*

For tabular RL with truncated Gaussian reward, the observation $o$ is a mixture of discrete random variables (i.e., the states and actions $s_h, a_h$) and continuous random variables (i.e., the rewards $r_h$). To prove Conditions 1-3, we deal with the discrete and continuous part of the observation separately. We also have flexibility in the reward distribution, e.g., our proof technique can deal with other distributions such as truncated Laplace distribution. We present these unified conditions for bounded random variables, but Conditions 2 and 3 do not hold for unbounded random variables because the base measure has an infinite volume (which contradicts to Condition 2), and the density of the observation cannot be lower bounded (which contradicts to Condition 3). However, we can still prove the same result using a slightly different approach (see Appendix E).

With Conditions 1-3, we can prove our main results for infinite hypothesis class. The asymptotic regret of Alg. 1 in this case still matches the lower bound exactly. The proof of Theorem 5.2 is deferred to Appendix C.1.

**Theorem 5.2.** *Suppose $\mathcal{F}$ is an infinite hypothesis class that satisfies Conditions 1-3, the regret of Alg. 1 satisfies $\limsup_{n\to\infty} \frac{\text{Reg}_{f^\star,n}}{\ln n} \leq \mathcal{C}(f^\star)$.*

As a corollary of Theorem 5.1 and Theorem 5.2, our algorithm strictly improves upon previous best gap-dependent bounds $\mathcal{O}(\sum_{s,a:\Delta(s,a)>0} 1/\Delta(s,a))$ in Xu et al. (2021) and Simchowitz & Jamieson (2019) because the gap-dependent bounds are not tight for many instances (Tirinzoni et al., 2021).

In the following we discuss the challenges when extending to infinite hypothesis settings.

**Covering Number.** With infinite hypothesis, we need to accept an accurate estimation even when there are infinitely many other choices. Recall that the accept event is

$$\mathcal{E}_{\text{acc}}^{\hat{f}} = \mathbb{I}\left[\forall g \in \mathcal{F} \text{ and } \pi^\star(g) \neq \pi^\star(\hat{f}), \sum_{i=1}^m \ln \frac{\hat{f}[\pi_i](o_i)}{g[\pi_i](o_i)} \geq \ln n\right]. \tag{12}$$

So informally speaking, we need to show that with high probability,

$$\forall g \in \mathcal{F} \text{ and } \pi^\star(g) \neq \pi^\star(\hat{f}), \quad \sum_{i=1}^m \ln \frac{\hat{f}[\pi_i](o_i)}{g[\pi_i](o_i)} \gtrsim (D_{1-\lambda}^w(\hat{f}\|g) - \epsilon)m \geq \ln n, \tag{13}$$

which is essentially an uniform concentration inequality as discussed in Section 4.3. So we resort to a covering number argument. The standard covering argument is not directly suitable in this case — even if $d(g, g')$ is small, it's still possible that $\ln(f[\pi](o)/g[\pi](o))$ is very different from $\ln(f[\pi](o)/g'[\pi](o))$ (especially when $g[\pi](o)$ is very close to 0). Instead, we consider the distribution with density $\hat{g}[\pi](o) \triangleq (g'[\pi](o) + \epsilon)/Z$ where $Z$ is the normalization factor, and only prove a one-sided covering (that is, $\ln(f[\pi](o)/g[\pi](o)) \geq \ln(f[\pi](o)/\hat{g}[\pi](o)) - \mathcal{O}(\epsilon)$). We state and prove the uniform concentration in Appendix F.

**Initialization.** With infinite hypothesis, we cannot hope to recover the true instance $f^\star$ exactly — some instances can be arbitrarily close to $f^\star$ and thus indistinguishable. Instead, we prove that the estimation in Step 1 satisfies $\sup_{\pi\in\Pi} D_{\text{KL}}(f^\star[\pi]\|\hat{f}[\pi]) \leq \text{poly}(\frac{\ln\ln n}{\ln n})$. The main lemma of Step 1 in the infinite hypothesis class case is stated in Lemma C.1.

# 6 CONCLUSION

In this paper, we design instance-optimal algorithms for general interactive decision making problems with finite decisions. As an instantiation, our algorithm is the first instance-optimal algorithm for tabular MDPs. For future works, we raise the following open questions.

(a) To implement Alg. 1, we need to solve $\mathcal{C}(f, (\ln\ln n)^{1/4}/2)$, which is a linear programming with $|\Pi|$ variables and infinitely many constraints. However, $|\Pi|$ is exponentially large for tabular MDPs. Can we compute this optimization problem efficiently for tabular MDPs?

(b) Although our algorithm is asymptotically optimal, the lower order terms may dominate the regret unless $n$ is very large. Can we design non-asymptotic instance optimal algorithms?

ACKNOWLEDGMENT

The authors would like to thank Yuanhao Wang, Jason D. Lee for helpful discussions, and the support from JD.com.

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
