# OpenReview forum: "Asymptotic Instance-Optimal Algorithms for Interactive Decision Making"
_ICLR.cc/2023/Conference — ICLR 2023 poster_

### Official Review · Reviewer_Mrke · 2022-10-24

**Confidence:** 3
**Correctness:** 3
**Technical Novelty And Significance:** 3
**Empirical Novelty And Significance:** Not applicable
**Recommendation:** 8

**Clarity, Quality, Novelty And Reproducibility:**

Clarity:

-Writing is mostly clear. Does the setup also capture undiscounted non-episodic RL tasks?

-Please provide more explanations of some assumptions made in the paper. For instance, why do you need to impose a unique optimal decision for each instance?

Quality and Novelty:

-Characterization of instance-dependent complexity of interactive decision-making problems, providing an asymptotic regret lower bounds for any consistent algorithm and proposing an algorithm that achieves asymptotic optimal regret are significant and novel.


**Strength And Weaknesses:**

Strengths:

-The paper proposes a framework for developing an instance-optimal algorithm that outperforms all consistent algorithms on every instance under mild conditions. The generality of the treatment in the paper, as it encompasses tabular reinforcement learning problems, is its main strength.

Weaknesses:

-No numerical results. It will be interesting to see how the algorithm performs with a finite horizon.

-Computational complexity of the algorithm can be prohibitive (this was mentioned as a limitation). More discussion on how it scales with the problem size and how to make it computationally efficient will greatly benefit the paper. Please comment on the computational complexity of different interactive decision-making problems.

-The algorithm requires the number of interactions as input.


**Summary Of The Paper:**

This paper proposes the first asymptotic instance-optimal algorithm for general interactive decision-making problems with a finite number of decisions. It provides an exact characterization of the complexity of each problem instance ${\cal C}(f)$, which is related to the amount of information to distinguish instance $f$ from other instances with different optimal decisions. The paper designs a three step algorithm based on uniform sampling (to construct MLE of f), identification (to boost confidence in the estimate of f) and exploitation (to maximize reward), whose lengths are carefully adjusted, and proves an asymptotic regret of ${\cal C}(f) \log n$ recovering well-known instance-dependent regret bounds for bandits and improving upon instance-dependent regret bounds for reinforcement learning.

**Summary Of The Review:**

Overall, this paper presents a solid theoretical contribution to interactive decision-making problems. The addition of insights about the finite-time performance and the practical applicability of the proposed algorithm will further strengthen the paper.

---

> ### Author Response · Authors · 2022-11-14
> **Response to Reviewer Mrke**
>
> > No numerical results. It will be interesting to see how the algorithm performs with a finite horizon.
>
> We thank the reviewer for the constructive comments. We leave the numerical evaluations to future works because our algorithm is designed to have a tight asymptotic regret bound, and its non-asymptotic regret could be very large even for a reasonable number of interactions. Hence, our algorithm may not have a dominant performance empirically unless $n$ is extremely large.
>
> Nonetheless, theoretically our dependence on the horizon for tabular RL problems should be asymptotically instance optimal.
>
> > Computational complexity of the algorithm can be prohibitive (this was mentioned as a limitation). More discussion on how it scales with the problem size and how to make it computationally efficient will greatly benefit the paper. Please comment on the computational complexity of different interactive decision-making problems.
>
> For certain specific interactive decision making problems, the optimization problem that defines $\mathcal{C}(f,n)$ can be solved efficiently. For example, the closed-form solution (i.e., the value of $w_\pi$) for multi-armed bandit problems is given in Eq. (85) in the Appendix.
>
> For tabular RL, the computation complexity of our algorithm is exponential in $S,A,H$ because it has to enumerate every instance. However, it is already highly non-trivial to prove an asymptotic instance-optimal regret bound regardless of the computation efficiency — the previous best works only provide results on bandit problems.
>
> > The algorithm requires the number of interactions as input.
>
> We thank the reviewer for this interesting question. The case with an unknown number of interactions is not the primary focus of this paper. Nonetheless, by the doubling trick, we can design an algorithm with asymptotic regret at most $4\mathcal{C}(f^\star)\ln n$ that does not require $n$ as input: suppose $\mathcal{A}(n)$ is our current T2C algorithm with input $n$, we can simply run $\mathcal{A}(2),\mathcal{A}(2^2),\mathcal{A}(2^4),\mathcal{A}(2^8),\cdots$ sequentially.
>
> In the following, we present the proof sketch for its asymptotic regret. Let $n_i=2^{2^i}$ be the length of the $i$-th epoch of the doubling. The total number of epochs will be $k=O(\ln\ln n)$. As a result, the regret introduced by Step 1 and Step 3 of the algorithm (repeated by $k$ times with length $n_1,n_2,\cdots,n_k$) is still $o(\ln n).$ The asymptotic regret of Step 2 is upper bounded by $\sum_{i=1}^{k}\ln n_i C(f^\star,(\ln\ln n_i)^{1/4})\le 2(\ln n)\sum_{i=1}^{k}2^{-(k-i)} C(f^\star,i^{1/4}).$ By the Silverman–Toeplitz theorem, $\lim_{k\to\infty}\sum_{i=1}^{k}2^{-(k-i)} C(f^\star,i^{1/4})=2C(f^\star)$. Therefore the asymptotic regret is upper bounded by $4C(f^\star)\ln n$.
>
> > Does the setup also capture undiscounted non-episodic RL tasks?
>
> The interactive decision making problem formulation does not capture undiscounted non-episodic RL tasks because the algorithm will not have the option to restart. We refer the reviewer to [1,2] for the instance-optimal rate for ergodic MDPs.
>
> [1] Burnetas, Apostolos N., and Michael N. Katehakis. Optimal adaptive policies for Markov decision processes.
> [2] Ok, Jungseul, Alexandre Proutiere, and Damianos Tranos. Exploration in structured reinforcement learning.
>
> > Please provide more explanations of some assumptions made in the paper. For instance, why do you need to impose a unique optimal decision for each instance?
>
> The unique optimal decision assumption seems to be a standard assumption in the literature for some technical reasons. Because the instances without unique optimal decisions is only a very small fraction or even has measure zero in generical, they intuitively shouldn't affect the instance optimally by much. That said, without uniqueness, we can still run our T2C algorithm but it’s unclear whether it is instance-optimal.
>
> When the optimal policy is not unique, we have to change the definition of our complexity measure $C(f,n)$ because the constraint $\pi^\star(g)\neq \pi^\star(f)$ in Eq. (3) is not rigorous when $\pi^\star(g)$ is a set instead of a single decision. One straightforward modification of Eq. (3) is to require that the instances $f,g$ are distinguishable if $\pi^\star(f)$ is not a subset of $\pi^\star(g)$. Then we can still prove the upper bound but the lower bound fails: imagine that $\mathcal{F}$ has only two instances but they share exactly one optimal decision, then the actual lower bound should be 0 because the optimal algorithm can simply output the shared optimal decision. However, the modified complexity measure is not zero.

---

> > ### Comment · Reviewer_Mrke · 2022-11-25
> > **Thanks for your reply**
> >
> > Please highlight the limitations and future research directions mentioned in your response in the final version of the paper.

---

### Official Review · Reviewer_ahML · 2022-10-24

**Confidence:** 3
**Correctness:** 4
**Technical Novelty And Significance:** 3
**Empirical Novelty And Significance:** 3
**Recommendation:** 8

**Clarity, Quality, Novelty And Reproducibility:**

The clarity, quality and novelty of the paper are very good. There is no experimental evaluation, so the reproducibility question is not relevant.

**Strength And Weaknesses:**

Strengths:
- interesting problem
- natural but elegant approach
- Well presented content


Weaknesses:
- the algorithm is not computable for every problem of interest, so it is unclear to me whether the results are also of practical relevance.


**Summary Of The Paper:**

The paper presents an asymptotic instance-optimal algorithm for decision problems under mild and natural conditions. This is interesting because it circumvents the weaknesses of minimax regret bounds. The idea behind the algorithm is that it alternates between having an estimate of the instance and testing whether estimate is correct. Doing this repeatedly results in increasing the confidence in the estimate. The optimality in the above statement, refers to comparing to the best \emph{consistent} algorithm for the instance at hand -- and consistent algorithms are very natural to consider for this purpose.


A typo: Page 2, belongs -> belong

**Summary Of The Review:**

This is a very interesting and good paper, I recommend acceptance.

---

> ### Author Response · Authors · 2022-11-14
> **Response to Reviewer ahML**
>
> We thank the reviewer for the comments, and for noting “this is a very interesting and good paper”.  In the following, we will address the reviewer’s comments in detail.
>
> > the algorithm is not computable for every problem of interest, so it is unclear to me whether the results are also of practical relevance.
>
> While we agree with the reviewer that the computation efficiency is one limitation of our paper, it is already highly non-trivial to prove an asymptotic instance-optimal regret bound even for tabular RL regardless of the computation efficiency — the previous best works only provide results on bandit problems. Therefore, we leave the practical relevance of our results to future works.

---

> > ### Comment · Reviewer_ahML · 2022-11-25
> > **Thanks**
> >
> > Thanks for the response.

---

### Official Review · Reviewer_hFWY · 2022-10-25

**Confidence:** 4
**Correctness:** 4
**Technical Novelty And Significance:** 4
**Empirical Novelty And Significance:** Not applicable
**Recommendation:** 10

**Clarity, Quality, Novelty And Reproducibility:**

Clarity is good. It is easy to follow this paper
This paper's conclusions are well supported by theoretical proofs.
The novelty is high, and the results are very good.
The paper does not have issues/concerns of reproducibility

**Strength And Weaknesses:**

Strength: This paper's results are very strong and impactful to the community. If the result is not wrong, then this should be the first instance-wise asymptotic algorithm for stochastic MAB, and closes the gap that has been studied for decades. The algorithm proposed in this paper uses a pretty novel method, and the linear-optimization based instance-wise factor is also new.

Weakness: No major weakness, but it is hard for me to guarantee that the proofs and mathematical conclusions are 100% correct.

**Summary Of The Paper:**

This paper studies the theoretical bounds of the classical stochastic MAB problems, and if the conclusion of this paper is not wrong, then as far as I know, this would be the first paper to close the gaps between the asymptotic instance-wise regret upper bounds and lower bounds.
To be specific, this paper proves that for any consistent algorithm A with regret R_A(n) (n is the time horizon), it must satisfy that limsup_{n->\inf}R_A(n)/log(n) >= C where C is a number determined by the instance and is independent of n and the algorithm. Then, this paper proposes an algorithm with regret bound R(n) such that limsup_{n->\inf}R(n)/log(n) <= C, which perfectly matches the upper bound with no gap.

**Summary Of The Review:**

This paper studies a very fundamental problem of stochastic MAB, and proves the asymptotic upper bounds and lower bounds for this problem, which closes the gap that has existed for decades. The algorithm and and conclusions in this paper are novel, and should inspire and influence all MAB related researches to help improve the theoretical conclusions on the upper bounds and lower bounds. Thus, I tend to give a strong accept.

---

> ### Author Response · Authors · 2022-11-14
> **Response to Reviewer hFWY**
>
> We thank the reviewer for their encouraging comments!

---

### Official Review · Reviewer_Li2S · 2022-10-25

**Confidence:** 5
**Clarity, Quality, Novelty And Reproducibility:** In Strength And Weaknesses
**Correctness:** 3
**Technical Novelty And Significance:** 3
**Empirical Novelty And Significance:** Not applicable
**Recommendation:** 8

**Strength And Weaknesses:**

This paper designs the first asymptotic instance-optimal algorithm for interactive decision making problems, while prior works mostly focus on the hardest instance. Overall, I think this paper is interesting and novel. However, due to my heavy review load, I feel so bad I can't go through all the details of this paper. Based on what I have read, I would highly recommend this paper and give it a score of 8.

In this part, I will list some questions about some details in this paper.
1. In the definition of $\mathcal{C}(f,n)$, the authors mention the last constraint (4) makes sure $w_{\pi}<\infty$ even if the decision has no regret. Could the authors be more specific in explaining how $w_{\pi}=\infty$ if we remove constraint (4)?
2. I'm a little confused with the instance complexity $\mathcal{C}(f)$ and some other instance complexity in [1,2]. Could the authors give some comparisons in rebuttal?
3. I understand this work provides a tight instance-optimal regret in the asymptotic regime. However, I'm more curious about how the technique used in this paper could inspire other future works.
4. About the assumption that the optimal policy is unique. I'm thinking about a simple multi-arm bandit problem, where $\mu_1=\mu_2>\mu_i$ for $i=3,...,n$. If the optimal policy is not unique, will the instance-optimality still hold? Or how will it fail?

Due to my limited time, I will consider adding more questions in the rebuttal phase.

[1]Koulik Khamaru, Eric Xia, Martin J Wainwright, and Michael I Jordan. Instance-optimality in optimal value estimation: Adaptivity via variance-reduced Q-learning. arXiv preprint arXiv:2106.14352,2021b.
[2]Li X, Yang W, Zhang Z, et al. Polyak-Ruppert Averaged Q-Leaning is Statistically Efficient[J]. arXiv preprint arXiv:2112.14582, 2021


**Summary Of The Paper:**

In Strength And Weaknesses

**Summary Of The Review:**

In Strength And Weaknesses

---

> ### Author Response · Authors · 2022-11-14
> **Response to Reviewer Li2S (part 1)**
>
> We thank the reviewer for the comments, and for noting “this paper is interesting and novel”. In the following, we will address the reviewer’s comments in detail.
>
> > In the definition of C(f,n), the authors mention the last constraint (4) makes sure $w_\pi<\infty$ even if the decision has no regret. Could the authors be more specific in explaining how $w_\pi=\infty$ if we remove constraint (4)?
>
> If we remove constraint (4), the optimal decision can be queried infinite number of times (i.e., $w_{\pi^\star}=\infty$) because $\pi^\star$ has no regret. When the hypothesis class is infinite, this constraint can fundamentally change the solution to $\mathcal{C}(f,n)$ in certain cases.
>
> For example, we can construct a MAB hypothesis class $\mathcal{F}=\\{\mu\in\mathbb{R}^{A}:\mu(1)\neq 0.5\\}\cup \{[0.5,0.1,\cdots,0.1]\}$ (where $\mu\in\mathbb{R}^{A}$ represents the mean reward of each arm). Then without the constraint $\|w_\pi\|\_{\infty}\le n$, the solution will be $w_1=\infty$ and $w_{i}=0,\forall i\neq 1$. In contrast, with $\|w_\pi\|_{\infty}\le n$, the solution should be $w_i>0,\forall i\neq 1$ (because there exits other instances in $\mathcal{F}$ whose mean reward of action 1 is arbitrarily close of 0.5).
>
> > I'm a little confused with the instance complexity C(f) and some other instance complexity in [1,2]. Could the authors give some comparisons in rebuttal?
>
> Comparing with the local minimax complexity [1]: The local minimax complexity measure, written in our notations, is:
> $$\mathfrak{M}_n(f)=\sup\_{f’\in\mathcal{F}}\inf\_{\text{algorithm}}\max\_{g\in \\{f,f’\\}}\text{Reg}\_{g,n}.$$ On a high level, it measures the regret of the hardest instance within $\{f,f’\}$ for any other instance $f’\in\mathcal{F}.$ For certain general decision making problems, the local minimax complexity can be strictly improved — there exists an algorithm that has a regret strictly smaller than $\mathfrak{M}_n(f)$ on some instance $f$, but also has regret at most $\mathfrak{M}_n(f’)$ on other instances $f’$.
> This is partly because when the hypothesis class $\mathcal{F}$ contrains only two instances, the local minimax complexity becomes the minimax complexity over $\mathcal{F}$. However, it is still possible that one of the instances in $\mathcal{F}$ is easier than the other.
>
> In the following, we construct such a general decision making problem. Suppose the hypothesis class is $\mathcal{F}={f_1,f_2}$ where $C(f_1)>C(f_2)$ (e.g., three-armed Bernoulli bandits with mean reward $[0,1,0.99]$ and $[1/2,0.99,1]$, respectively. In this case $C(f_2)=0$ but $C(f_1)>0$ because $KL(Ber(1/2)||Ber(0))=\infty,KL(Ber(0)||Ber(1/2))<\infty$.) Then the local minimax complexity is at least $\mathfrak{M}_n(f_1)=\mathfrak{M}_n(f_2)\ge \max(C(f_1)\ln n, C(f_2)\ln n)=C(f_2)\ln n$ for large enough $n$. This is because (1) any consistent algorithm will have regret at least $C(f_1)\ln n$ on $f_1$ and $C(f_2)\ln n$ on $f_2$, and (2) any inconsistent algorithm will have $n^p$ regret on some instance in $\mathcal{F}$ for some constant $p>0$. However, our instance optimal algorithm has a strictly smaller regret on $f_2$ because $C(f_2)<C(f_1)=\mathfrak{M}_n(f_2)$, while having a regret that matches $\mathfrak{M}_n(f_1)$ on $f_1$.
>
> Comparing with the Cramer-Rao-type lower bound [2]: On a high level, the Cramer-Rao lower bound in [2] applies to parameter estimation problems (e.g., estimating the Q-function), and our complexity measure applies to regret minimization for general decision making problems. Our complexity is not comparable with [2] in general — the best parameter estimation algorithm is not optimal for regret minimization because estimating the Q-function is not necessary for finding the optimal policy, and vice versa.
>
> Technically, [2] focus on the RL with generative model — on every round the algorithm receives one sample from *every* state-action pairs. In other words, the algorithm does not need to explore and the question becomes more like supervised learning. Hence, [2] can prove the Cramer-Rao-type lower bound for regular estimators.
>
> > I understand this work provides a tight instance-optimal regret in the asymptotic regime. However, I'm more curious about how the technique used in this paper could inspire other future works.
>
> On a high level, we essentially reduce asymptotic instance-optimal regret minimization to optimal hypothesis testing with active data collection. We believe that this reduction can be also applied to other questions such as instance-optimal PAC bounds, instance-optimal offline RL, etc. Potentially, designing a computationally efficient version of our algorithm may also lead to new practical algorithms.

---

> ### Author Response · Authors · 2022-11-14
> **Response to Reviewer Li2S (part 2)**
>
> > About the assumption that the optimal policy is unique. I'm thinking about a simple multi-arm bandit problem, where $\mu_1=\mu_2>\mu_i$ for $i=3,...,n$. If the optimal policy is not unique, will the instance-optimality still hold? Or how will it fail?
>
> When the optimal policy is not unique, we have to change the definition of our complexity measure $C(f,n)$ because the constraint $\pi^\star(g)\neq \pi^\star(f)$ in Eq. (3) is not rigorous when $\pi^\star(g)$ is a set instead of a single decision. Either the lower bound or the upper bound will fail depending on how to modify Eq. (3).
>
> One straightforward modification of Eq. (3) is to require that the instances $f,g$ are distinguishable if $\pi^\star(f)$ is not a subset of $\pi^\star(g)$. In this case, we can still prove the upper bound but the lower bound fails: imagine that $\mathcal{F}$ has only two instances but they share exactly one optimal decision, then the actual lower bound should be 0 because the optimal algorithm can simply output the shared optimal decision. However, the modified complexity measure is not zero.
>
> We can also require that the instances $f,g$ are distinguishable if $\pi^\star(g)$ and $\pi^\star(f)$ do not have any intersection. In this case the lower bound still holds but the upper bound fails: in step 3, it’s unclear which optimal decision should the algorithm commits to.

---

### Official Review · Reviewer_sYY8 · 2022-10-27

**Confidence:** 4
**Correctness:** 4
**Technical Novelty And Significance:** 3
**Empirical Novelty And Significance:** Not applicable
**Recommendation:** 6

**Clarity, Quality, Novelty And Reproducibility:**

The paper is well written and the results are clearly presented. The body includes a proof sketch of the main result which greatly helps the reader. The main proof idea is solid; however I have not checked the appendix in detail.

Prior work is adequately discussed, however misses the important reference of [Komiyama et al; 2015]. In fact this previous paper encompasses the proposed setting as long as the observation space is finite. It would be great if the authors could add a discussion of this paper.

Komiyama, Junpei, Junya Honda, and Hiroshi Nakagawa. "Regret lower bound and optimal algorithm in finite stochastic partial monitoring." Advances in Neural Information Processing Systems 28 (2015).

**Strength And Weaknesses:**

The main contribution of the paper is the analysis of asymptotic optimality in a fairly general setting. The results require only mild conditions on the observation distributions, and the usual caveats of a finite action/decision space and a unique optimal action. At least in this context, I believe that the analysis of the log-likelihood ratio test is novel (Lemma 4.3 and its use in Theorem 3.3). The paper also discusses the required covering arguments to extend the result to infinite hpyothesis classes

On the other hand, the algorithmic approach is the same as in Lattimore et al (2017). The algorithm is neither practical from a sample complexity perspective (exponential lower order terms, horizon dependent, not worst-case optimal); nor computationally feasible unless both the hypothesis space and decision set is small. The result should therefore be understand a proof of concept with regard to the achievable sample complexity. It should also noted that going beyond this proof of concept is likely to require significantly further insights and assumptions.

**Summary Of The Paper:**

This paper studies asymptotic instance optimality for interactive decision making with finite decision/action set. The model includes bandits and reinforcement learning, as long as the reward is a function of the observation.

The authors show that the standard, asymptotic lower bound by Lai and Robbins applies to this setting. They further adapt the approach by Lattimore et al (2017) to the more general setting to match the lower bound (asymptotically). Analysis is presented for both finite hypothesis classes and infinite hypothesis classes.

**Summary Of The Review:**

In summary, the conceptual contributions of this paper are somewhat limited as they more or less directly extend prior work; however the technical contributions due to the generality of the setting are non-trivial and could inspire follow up work.

---

> ### Author Response · Authors · 2022-11-14
> **Response to Reviewer sYY8**
>
> We thank the reviewer for the comments, and for noting our “analysis of the log-likelihood ratio test is novel”. In the following, we will address the reviewer’s comments in detail.
>
> > the algorithmic approach is the same as in Lattimore et al (2017).
>
> While the three-step algorithmic framework is the same as Lattimore et al (2017) on a very high level (and we have acknowledged this similarity in the paper), our T2C algorithm relies on its key second step which requires fundamentally novel analysis: the algorithm in Lattimore et al (2017) only uses the reward information, whereas the observation for general interactive decision making problem contains much more information such as the transition in tabular RL problems. Hence, to generalize to interactive decision making problems, one of our contributions is to deal with all types of observations in an abstract way using the log likelihood ratio hypothesis testing method. Conceptually, it was even unclear whether instance-optimal algoirthms exsit for general interactive decision making problems (such as RL), and our results provide the first positive answer to this question.
>
> > The algorithm is neither practical from a sample complexity perspective  (exponential lower order terms, horizon dependent, not worst-case optimal); nor computationally feasible … The result should therefore be understand a proof of concept with regard to the achievable sample complexity.
>
> While studying computation efficiency and non-asymptotic regret for general decision making problems are very interesting and important directions (and we thank the reviewer again for these constructive comments), it was already highly non-trivial to prove an asymptotic instance-optimal regret bound for RL regardless of the computation efficiency — the previous best works only provide results on bandit problems. Therefore, we leave these questions to future works.
>
> > Prior work is adequately discussed, however misses the important reference of [Komiyama et al; 2015]. In fact this previous paper encompasses the proposed setting as long as the observation space is finite.
>
> We thank the reviewer for pointing out additional related works, and we will update the paper accordingly upon revision. We agree with the reviewer that the algorithm in Komiyama et al. (2015) is applicable to *finite* observation space, which is a special case of the interactive decision making problem (that *does not* include standard multi-armed bandits with Gaussian noise). We also note that a critical step in the analysis of Komiyama et al. (2015) (Eq. (22) of [1]) can only handle finite observation space, and extending this step to continuous observation space requires novel analysis.
>
> [1] Komiyama, Junpei, Junya Honda, and Hiroshi Nakagawa. "Regret lower bound and optimal algorithm in finite stochastic partial monitoring." https://arxiv.org/pdf/1509.09011.pdf

---

### Decision · Program_Chairs · 2023-01-20

**Decision:**

Accept: poster

**Justification For Why Not Higher Score:**

While most reviewers are quite enthusiastic, I see two limitations of the paper:
- The model is not as general as suggested in the title (e.g., it does not apply to non-episodic RL).
- While the idea of using MLE estimation and the likelihood ratio test is novel in this context, the extension on top of the work of Lattimore & Szepesvari (2017) is somewhat limited.
- The algorithm is not practical in the general case.

**Justification For Why Not Lower Score:**

The results are novel and of interest to the community working on sequential decision making problems.

**Metareview: Summary, Strengths And Weaknesses:**

This paper studies asymptotic instance optimality for interactive decision making with finite decision/action set. The model includes bandits and episodic reinforcement learning, as long as the reward is a function of the observation. The authors show that the standard, asymptotic lower bound applies to this setting. They further adapt the approach by Lattimore & Szepesvari (2017) to the more general setting to match the lower bound (asymptotically). Analysis is presented for both finite hypothesis classes and infinite hypothesis classes, under some mild assumptions. The main improvement over the work of Lattimore & Szepesvari is the application of a likelihood ratio test about the most likely hypothesis, which is necessary to achieve the generality presented in the paper.

The paper contains a number of typos, so the authors are encouraged to carefully check and correct these for the final version.

**Note From Pc:**

if the above contains the word "oral" or "spotlight" please see: "oral" presentation means -> notable-top-5% and "spotlight" means -> notable-top-25%. As stated in our emails, we are disassociating presentation type from AC recommendations